# Analysis of Data-Derived SeaWinds Normalized Radar Cross-Section Noise

Giuseppe Grieco [1,*], Ad Stoffelen [2], Anton Verhoef [2], Jur Vogelzang [2] and Marcos Portabella [3]

[1] Istituto di Scienze Marine, Consiglio Nazionale delle Ricerche (ISMAR-CNR), Calata Porta di Massa, 80133 Napoli, Italy

[2] Koninklijk Nederlands Meteorologisch Instituut (KNMI), Utrechtseweg 297, 3731 GA De Bilt, The Netherlands

[3] Institut de Ciències del Mar, Consejo Superior de Investigaciones Cientificas (ICM-CSIC), Passeig Marítim de la Barceloneta, 37, 08003 Barcelona, Spain

* Correspondence: giuseppe.grieco@cnr.it

**Abstract:** The normalized standard deviation ($K_p$) of the noise that affects scatterometer Normalized Radar Cross-Sections ($\sigma_0$s) plays a key role in the ocean and more in particular coastal wind retrieval procedures and the a posteriori quality control. This paper presents a method based on SeaWinds measurements to estimate $K_p$s. The method computes the standard deviation of the differences between the full-resolution (slice) $\sigma_0$s and the footprint (egg) $\sigma_0$. The results are compared to the median of $K_p$s provided with SeaWinds $\sigma_0$s, showing some non-negligible differences. $K_p$s estimated on non-homogeneous surfaces are larger than those estimated on sea, whereas no differences are appreciated in the provided $K_p$s, which is likely due to the ability of this methodology to account for the effect of the scene variability in the estimates. The presence of inter-slice biases is demonstrated with a trend with the antenna azimuth angle. A multi-collocation slice cross-calibration procedure is suggested for the retrieval stage. Finally, a theoretical model of the distribution of $\sigma_0$s is proposed and used to validate $K_p$s. The results prove the efficacy of this model and that the provided $K_p$s seem to be largely underestimated at low-wind regimes.

**Keywords:** SeaWinds; normalized radar cross-section noise; wind




## 1. Introduction

Wind is the main ocean forcing of circulation processes at small to medium spatiotemporal scales. Therefore, highly sampled and accurate ocean winds are of paramount importance for various scientific and civil applications.

Scatterometers represent the gold standard for monitoring the ocean wind vector field. SeaWinds is a rotating pencil beam scatterometer that flew aboard the US polar orbiting satellite platform Quik Scatterometer (QuikSCAT) from July 1999 until November 2009 as well as aboard the polar orbiting Japanese satellite Advanced Earth Observing Satellite 2 (ADEOS II), even if for only ten months [1]. SeaWinds measures the backscattered normalized radar cross-section ($\sigma_0$), which is defined as the ratio of the power scattered back to the radar receiver over the incident radar power density per unit of solid angle on the target and per unit of target area, as if the radiation was isotropic. SeaWinds architecture has been replicated for scatterometers of several subsequent satellite missions, among which are the Chinese family Haiyang-2 (HY-2) and the Indian Oceansat 2 and 3 and Scatterometer Satellite 1 (Scatsat 1). It consists of a rotating parabolic antenna that sends horizontally polarized (HH-Pol) and vertically polarized (VV-Pol) signals at incidence angles of 46° and 54°, respectively. The antenna radiates microwave pulses with a carrier frequency of 13.4 GHz (Ku-band), covering a swath of 1800 km centered around the spacecraft's nadir. Its design parameters ensure that every point on the sea surface is covered by four different combinations *pol; view* (flavors), namely: HH-Pol, fore; HH-Pol, aft; VV-Pol, fore; and VV-Pol, aft. Using range filtering, each SeaWinds footprint

(egg) is resolved into eight slices, whose linear dimensions are approximately $24 \times 4$ km$^2$. Figure 1 represents the geometry of an anticlockwise conically scanning pencil-beam of a scatterometer such as SeaWinds. SeaWinds covered 90% of the globe every day. The stress-equivalent 10 m SeaWinds-derived wind vector data records are available on a regular Wind Vector Cell (WVC) grid with a spacing of 25 and 50 km and can be freely downloaded from the Ocean Sea Ice Satellite Application Facility (OSI SAF) website [2,3]. Alternatively, one can download 12.5, 25 and 50 km data from the Physical Oceanography Distributed Active Archive Center (PODAAC) website [4]. Full-resolution (FR) Level 1 files are also available from the same website.

It is well known that near-coastal acquisitions from scatterometers suffer from land contamination, which can lead to biased retrieved winds. To address this problem, Ref. [5] discarded all slices with a Land Contribution Ratio (LCR) greater than a given threshold, which was typically set at 2%. LCR is a weighted average of a high-resolution Land–Sea Mask (LSM) by means of the Spatial Response Function (SRF) of the slice. OSI SAF is developing a SeaWinds-derived coastal product in a fashion similar to that of the Advanced Scatterometer (ASCAT) [6]. Ref. [6] were inspired by [5,7] but by applying a correction to the contaminated measurements rather than by discarding them. This approach proved to be effective in improving both coastal sampling and accuracy, and it will also be considered for SeaWinds in the upcoming future.

Noise information is fundamental in the inverse problem [8] and hence in the retrieval of the wind field from a scatterometer $\sigma_0$s [9]. In fact, noise is informative of the accuracy of the observations and impacts the probability distribution function of the retrievals. In the case of active microwave sensors, this information is provided in the normalized standard deviation of $\sigma_0$ noise ($K_p$), which is defined as the ratio of the standard deviation of $\sigma_0$ to its expected value [10]. The expected value of $\sigma_0$ is obtained by subtracting the estimate of the thermal noise that affects $\sigma_0$ from the signal-plus-noise $\sigma_0$ measurement. $K_p$ takes into account thermal noise and the fading effect [11], and for scatterometers, it is also affected by the variability of the backscatter field in the footprint [12].

$K_p$ is estimated using a second-order polynomial of the inverse of the signal-to-noise ratio (SNR) [10], which does not account for the effect of scene variability. Unfortunately, in the case of Seawinds, $K_p$ is not provided with additional information related to its uncertainty. SeaWinds $\sigma_0$s are very noisy, and the variability of $K_p$ is large in low-wind regimes ($\leq 5$ ms$^{-1}$). Note that to mitigate the effect of the variability of $K_p$, an additional logarithmic term based on the variance of $\sigma_0$ is often added to the cost function in the operational retrieval algorithm of QuikSCAT winds [13]. $K_p$ can also be used for Quality Control (QC); therefore, its variability is also important for this purpose. In [14], the authors show how SeaWinds $K_p$s are used to calculate the so-called normalized (inversion) residual index to detect rain contamination in the retrieved winds.

This paper presents an empirical method to accurately estimate $K_p$ from the data acquired by SeaWinds. This method is also suitable for all scatterometers such as SeaWinds. These estimates are then compared to the median values of the $K_p$s provided in the FR files (hereafter referred to as product $K_p$s). Furthermore, the variability of $K_p$ relative to the kind of surface (all types of surface or sea only), the wind regimes and the polarization of the carrier signal are assessed. The dependence of any bias on the antenna azimuth angle is also analyzed, together with the dependence of any bias on the intra-egg variability of the incidence angle. The impact of such biases on the estimate of $K_p$ is also discussed. Finally, the empirical estimates of $K_p$ are validated by comparing the distributions of real values of $\sigma_0$s with those obtained with the estimates of $K_p$.

A proper inter-calibration procedure aiming at removing the intra-egg biases and more accurate empirical $K_p$s may be beneficial for wind retrievals, especially in coastal areas, even if this analysis is left for the future.

The paper is organized as follows: Section 2 describes the dataset used in this study and the methodology applied. In particular, the data and the QC scheme applied are described in Section 2.1; the methodology used to estimate $K_p$s from the data is described

in Section 2.2.1, the impact of intra-egg $\sigma_0$ biases on the estimates of $K_p$ in Section 2.2.2 and the analytical distribution of $\sigma_0$ is outlined in Section 2.2.3; the results are shown and discussed in Section 3, while the conclusions are given in Section 4.

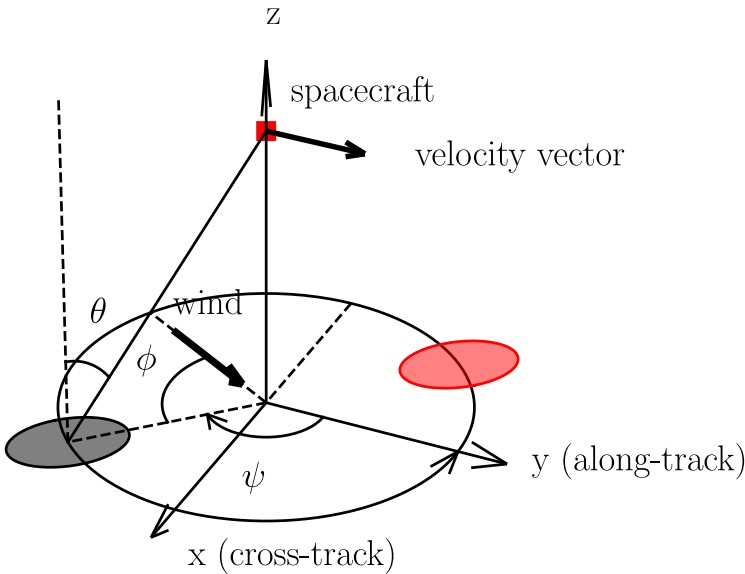

**Figure 1.** Schematic representation of the geomtery of a conically scanning spaceborne pencil-beam scatterometer in a reference system centered around the spacecraft's nadir. The spacecraft is represented by the red square, and the red (gray) ellipse represents the antenna footprint in acquisition mode fore (aft). $\psi$ represents the antenna azimuth angle, $\phi$ the relative wind direction and $\theta$ is the incidence angle.

## 2. Data and Methods

### 2.1. Data

Fourteen FR files dated 10 April 2007 have been used in this study, each corresponding to one entire orbit. Their orbit numbers range from 40,651 to 40,664. As mentioned above, they can be downloaded from the PODAAC website [4]. The entire content of each FR file is well described in the SeaWinds reference manual [13]. In the following, only the most relevant information for this study is reported and discussed. All fields used in this study are reported in Table 1. The acronyms "sc" and "ECEF" stand for "spacecraft" and "Earth-Centered Earth-Fixed", respectively, while the prefix `cell` refers to the entire scatterometer egg. All arrays are provided in a time-ordered fashion, following the beam rotation. `n_f`, `n_p`, and `n_s` stand for "number of telemetry frames", "number of microwave pulses", and "number of slices", respectively. `n_f` may vary with the orbit, where its typical value is around 11,240, corresponding to about 100 min or one complete satellite revolution. `n_p` is the number of electromagnetic pulses that the antenna sends, is equal to 100 and covers one-sixth of an antenna rotation. This implies that the pulse-to-pulse distance on the earth surface is 10 km for the outer beam and 7.8 km for the inner beam. `n_s` is 8, cutting a cell (egg) $\sigma_0$ SeaWinds footprint into 8 $\sigma_0$ values, which are spatially separated by about 4 km across the range-looking direction. In fact, thanks to range filtering, the SeaWinds footprint is resolved into eight different measurements, whose shape suggests the name "slice" (see Figure 2).

**Table 1.** List of fields used in this study. Under "shape", n_f, n_p and n_s stand for "number of frames", "number of pulses" and "number of slices", respectively. n_p is equal to 100 and n_s to 8, while n_f may vary with the orbit. Its typical value is around 11,240. The acronyms "sc" and "ECEF" stand for "spacecraft" and "Earth-Centered Earth-Fixed", respectively.

| Field | Shape | Full Name |
|---|---|---|
| sc_lat | (n_f,n_p,n_s) | sc latitude |
| sc_lon | (n_f,n_p,n_s) | sc longitude |
| sc_alt | (n_f,n_p,n_s) | sc altitude |
| x_pos | (n_f,n_p,n_s) | sc x-position ECEF |
| y_pos | (n_f,n_p,n_s) | sc y-position ECEF |
| z_pos | (n_f,n_p,n_s) | sc z-position ECEF |
| x_vel | (n_f,n_p,n_s) | sc x-velocity ECEF |
| y_vel | (n_f,n_p,n_s) | sc y-velocity ECEF |
| z_vel | (n_f,n_p,n_s) | sc z-velocity ECEF |
| cell_lat | (n_f,n_p) | cell latitude |
| cell_lon | (n_f,n_p) | cell longitude |
| cell_sigma0 | (n_f,n_p) | cell $\sigma_0$ |
| cell_azimuth | (n_f,n_p) | cell azimuth |
| cell_incidence | (n_f,n_p) | cell incidence angle |
| ant_azimuth | (n_f,n_p) | antenna azimuth |
| slice_snr | (n_f,n_p,n_s) | slice SNR |
| slice_kpc_a | (n_f,n_p,n_s) | slice kp a coefficient |
| slice_kpc_b | 1 | slice kp b coefficient |
| slice_kpc_c | 1 | slice kp c coefficient |
| slice_azimuth | (n_f,n_p,n_s) | slice azimuth |
| slice_incidence | (n_f,n_p,n_s) | slice incidence angle |
| slice_sigma0 | (n_f,n_p,n_s) | slice $\sigma_0$ |
| orbit_time | (n_f) | orbit time |
| **Quality flag** | **Shape** | **Full name** |
| frame_err_status | (n_f) | frame error status |
| frame_inst_status | (n_f) | frame instrument status |
| frame_qual_flag | (n_f) | frame quality flag |
| sigma0_mode_flag | (n_f,n_p) | $\sigma_0$ mode flag |
| sigma0_qual_flag | (n_f,n_p) | $\sigma_0$ quality flag |
| slice_qual_flag | (n_f,n_p) | slice quality flag |
| **File attributes** | | |
| EquatorCrossingLongitude | 1 | Equator Crossing Longitude |
| orbit_inclination | 1 | Orbit inclination |
| rev_orbit_period | 1 | Orbit revolution period |
| orbit_semi_major_axis | 1 | Orbit semi-major axis |
| orbit_eccentricity | 1 | Orbit eccentricity |

Quality Control (QC)

All data have been QCed before use in order to discard unreliable measurements. For the sake of reproducibility, the QC scheme is reported here:

- `frame_err_status` is required to be 0. This requirement ensures that neither an unusual instrument condition applies, nor bad ephemeris, nor bad attitude.
- bit 4 (0-based) of `frame_qual_flag` is required to be 0; otherwise, bad data are present in the frame.
- bits 0–3 of `frame_inst_status` are required to be 0, while bits 4–6 are required to be "011". Bits 0–1 account for the "Current Mode", which can be "Wind Observation Mode", "Calibration Mode", "Standby Mode" or "Receive Only Mode". Bit 2 accounts for the "First Pulse Count in the Frame" (Pulse A first or Pulse B first) and bit 3

accounts for the "Antenna Spin Rate" (Nominal or Alternate Rate). Finally, bits 4–6 account for the "Slice Resolution Mode".

- bits 0, 4–9 of `sigma0_qual_flag` are required to be 0. This condition ensures that
  - The egg is usable (bit 0);
  - The scatterometer pulse is acceptable (bit 4);
  - The $\sigma_0$ cell location algorithm converges (bit 5);
  - The frequency shift is within the range of the x factor table (bit 6);
  - The spacecraft temperature is within the calibration coefficient range (bit 7);
  - An applicable attitude record was found for this $\sigma_0$ (bit 8);
  - Interpolated ephemeris data are acceptable for this $\sigma_0$ (bit 9).

Bits 1, 2, and 3 account for the SNR level, the sign of $\sigma_0$, and the admitted range of $\sigma_0$s. None of these constraints is applied to the noise estimation of the slice $\sigma_0$. In fact, a fair estimate of the noise of the slice $\sigma_0$ must take into account both low SNR measurements, negative values and $\sigma_0$s outside of the expected range; otherwise, the distribution of $\sigma_0$ could be truncated and some artificial biases could be introduced.

QC filters out ≈0.6% of HH-Pol acquisitions and ≈0.7% of those VV-Pol.

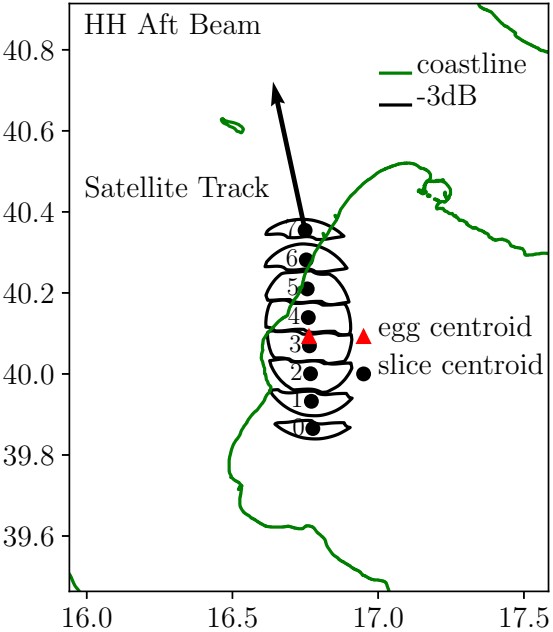

**Figure 2.** Example of SeaWinds inner-beam aft egg in a coastal region of southern Italy, in the Mediterranean basin, acquired with an antenna azimuth angle equal to ≈190°. The −3 dB contours of the slice spatial response function are depicted with black solid lines. The slice index of each slice is reported into the curves on a 0-based numbering. The string HH stands for HH-Pol.

### 2.2. Methods

#### 2.2.1. Estimate of the Expected $K_p$ ($K_p^{EMP}$)

The main objective of this study is to evaluate the noise from the slice $\sigma_0$. To pursue this aim, all egg data are grouped in 1 dB width bins around five different levels of $\sigma_0$, corresponding to the low-to-medium and high-wind speed regimes. In practice, for each of the acquisition polarizations, five reference levels of $\sigma_0$ are calculated by averaging the NSCAT4DS Geophysical Model Function (GMF) [15] corresponding to the wind speed values ranging from 5 to 15 ms$^{-1}$ with steps of 2.5 ms$^{-1}$ throughout the entire range of relative azimuth angles, according to the formula depicted in Equation (1).

$$\sigma_0^{U_T} \triangleq \frac{1}{2\pi} \int_0^{2\pi} NSCAT4DS(U_T, \phi, \theta, p) d\phi \tag{1}$$

where $U_T$ is the desired target wind speed, $\phi$ is the wind direction relative to the antenna, $\theta$ is the incidence angle and $p$ is the beam polarization (HH-Pol or VV-Pol). For the sake of completeness, we recall here that the incidence angle of each polarization beam is almost constant, as is expected for a pencil beam scatterometer such as SeaWinds. The inner beam (HH-Pol) incidence angle is equal to 46.2° (±0.7°), while that of the outer beam (VV-Pol) is 54.0° (±0.6°). These are the values that have been used in Equation (1). It is important to note that the NSCAT4DS GMF has been used only for this purpose.

All measurements are segregated according to:

- The antenna beam polarization;
- The view (fore or aft);
- The slice index.

Then, the operative definition of $K_p$ is applied, which reads as follows:

$$\hat{K}_p^{EMP,i} \triangleq \frac{\sqrt{E[(\hat{\sigma}_{0i} - \bar{\sigma}_{0i})^2]}}{\bar{\sigma}_{0i}} \tag{2}$$

where $EMP$ stands for empirical and $\bar{\sigma}_{0i}$ is the expected value for slice index $i$. This value is set to the egg $\sigma_0$. By definition, $K_p^{EMP}$ is representative of the noise associated with a set of measurements, as it is computed with Equation (2) over $N > 1$ samples. It is not possible to associate any $K_p^{EMP}$ with any single measurement. Therefore, $K_p^{EMP}$ is compared to the median value of product $K_p$s ($K_p^{MED}$). We have chosen the median instead of the average because the distribution of $K_p$ is not expected to be symmetric; therefore, the median may be more representative than the average. The median is computed for the same data set. Then, a sensitivity analysis is performed to assess the presence of any biases between the slices, the presence of any bias trends with respect to the antenna azimuth angle and the presence of any differences between the estimates of $K_p$ on the sea and those on all other types of surfaces.

Finally, the random distribution of $\hat{\sigma}_0$s obtained by applying the theoretical model derived from [10] (Section 2.2.3) using $K_p^{EMP}$ is compared to the distribution obtained using product $K_p$s and $K_p^{MED}$ to the distribution of real measurements in the samples.

### 2.2.2. Impact of the Intra-Egg Bias on $K_p^{EMP}$

The effective $\sigma_0$ bias induced by the difference in the incidence angle between the slice and the egg centroids depends on the wind direction. This information is, of course, not available. However, we can estimate the bias in the worst-case scenario (WCS), using the GMF as in Equation (3), where $\theta$ is the incidence angle, $\phi$ is the wind direction relative to the antenna beam, $U$ is the wind speed, $N4DS$ stands for the GMF NSCAT4DS, $i$ is the slice index, $j$ is the sample index and $sgn(i)$ equals $-1$ for the slices with indices up to 4 (the farthest ones) and 1 for the remaining slices (see Figure 2). In fact, the GMF is a monotonically decreasing function of $\theta$. That is, if $\theta_{ij}$ is higher (lower) than $\theta_{egg,j}$ as for the first four slices (last four), $\sigma_{0,i}$ is expected to be lower (higher) than $\sigma_{0,egg}$, if we accept that the geophysical variability in the egg is negligible. Figure 2 shows the inner beam $-3$ dB contours of the spatial response function (SRF) of the slices in a region in southern Italy, together with some additional information that can help better understand the acquisition geometry of SeaWinds. Slice indices are reported in the contours with a 0-based numbering, while the satellite track is depicted with a black arrow centered on the 8th slice to emphasize that the antenna azimuth angle is around 190°. Finally, the slice centroids are represented by black dots, whereas the egg centroid is represented by a red triangle.

WCS means that we consider the relative wind direction that produces the largest absolute deviation from $\sigma_{0,egg}$. The solid line of Figure 3 shows the expected value of the deviation of $\sigma_0$ from $\sigma_{0,egg}$ in linear units (LU), for the inner aft slice with index 0 for a wind speed equal to 5 ms$^{-1}$, as predicted by NSCAT4DS, throughout the entire orbit with number 40,651. The average value of the solid line is represented by the dashed line. Note that all the values are negative, which is consistent with what is expected. In this case, the

WCS is represented by the deviation at $\phi = 5°$, which is much larger (in absolute value) than the average value. This value can reasonably be considered to be the upper limit of the bias in this specific case. The maximum bias evaluated throughout the orbit is considered (Equation (3)). The ratio of this value to the expected value of $\sigma_0$ gives the upper limit of the impact of such a bias on $K_p^{EMP}$ in the same units.

$$b_i = sgn(i)max_j[max_\phi|N4DS(U,\theta_{ij},\phi) - N4DS(U,\theta_{egg,j},\phi)|]$$
$$\forall i \in 0,\ldots,7$$
$$\forall j \in 1,\ldots,N \qquad (3)$$
$$sgn(i) = -1 \forall i \in 0,\ldots,3$$
$$sgn(i) = 1 \forall i \in 4,\ldots,7$$

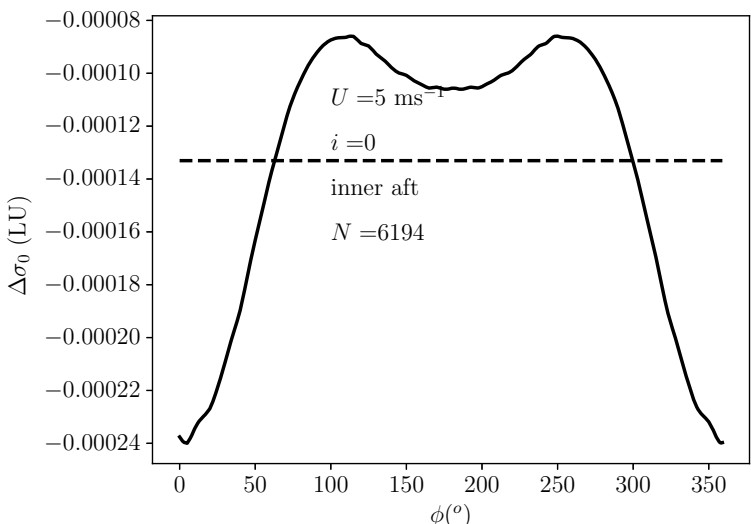

**Figure 3.** Solid line: expected $\sigma_0$ deviations in linear units (LU) from $\sigma_{0,egg}$ for the inner (HH-Pol) aft slice with index 0 as a function of the relative wind direction ($\phi$) simulated with NSCAT4DS at a wind speed equal to $5\,\text{ms}^{-1}$. Dashed line: average value of the solid line. $N$ represents the number of samples used in this simulation. The data used are from the orbit with number 40,651.

### 2.2.3. Theoretical Model of the Distribution of $\sigma_0$

From a physical point of view, $\sigma_0$ represents a normalized radar cross-section and, therefore, is positive definite. $\sigma_0$ is proportional to the power of the signal that reaches the receiver antenna, which is, in turn, equal to the sum of the squares of the real and imaginary parts of the signal voltage. Therefore, in the case of only one look, a noise-free $\sigma_0$ distribution would be related to a $\chi^2$ distribution with two degrees of freedom (dof), assuming that the real and imaginary parts of the signal are Gaussian distributed with identical standard deviation (equal to the amplitude of the voltage signal) and null mean [10]. In the case of a higher number of looks, $\sigma_0$ is distributed as normalized $\chi^2$, as described in Equation (4)

$$pdf_{\sigma_0} = \frac{1}{\alpha_{\sigma_0}2^{\frac{2}{k}}\Gamma(\frac{k}{2})}\left(\frac{\sigma_0}{\alpha_{\sigma_0}}\right)^{\frac{k}{2}-1}\exp{-\frac{\sigma_0}{2\alpha_{\sigma_0}}}$$
$$\alpha_{\sigma_0} = \frac{\mu_{\sigma_0}}{k} \qquad (4)$$
$$\sigma_{\sigma_0} = \mu_{\sigma_0}\sqrt{\frac{2}{k}}$$
$$k = \frac{2}{K_p^2}$$

where $k$ stands for dof, $\mu$ stands for expected value, and $\sigma$ stands for standard deviation. Once $K_p$ and $\mu_{\sigma_0}$ are known, $k$ and $\alpha_{\sigma_0}$ can be calculated. With the help of a random generator tool and the noise model described in [1], the distribution of $\sigma_0$s can be easily simulated. In this study, the random generator package `random` of the `python` library `numpy` has been used, as shown in the following formula:

$$\hat{\bar{\sigma}}_0 = \frac{1}{k}\texttt{numpy.random.chisquare}(k, n)\bar{\sigma}_0 \tag{5}$$

with $n$ being the number of samples and $\bar{\sigma}_0$ being the expected value of $\sigma_0$, and the symbol $\check{}$ indicates that the output of Equation (5) consists of $n \geq 1$ elements.

The $\sigma_0$ distribution can be obtained by simulating the noise-equivalent $\sigma_0$ ($\sigma_0^N$) and the $\sigma_0$ of the signal plus noise ($\sigma_0^{S+N}$) separately (using Equation (5)), and then, the former is subtracted from the latter, as happens in the SeaWinds signal processing chain [16]. In fact, both follow the same analytical distribution of Equation (4) and can be simulated using Equation (5), provided that the appropriate parameters are used. In this implementation, $\mu_{\sigma_0^N}$ is calculated by dividing $\sigma_{0,egg}$ by $SNR$, for each realization, while its standard deviation is obtained using Equation (40) of [1]. All the parameters used in that equation are available in the cited paper. They are not reported here for the sake of brevity. $\sigma_{\sigma_0^{S+N}}$ is computed using Equation (6), which reads

$$\sigma_{\sigma_{0,ij}^{S+N}} = \sqrt{(K_{p,ij}\sigma_{0,ij})^2 - \sigma_{\sigma_{0,ij}^N}^2} \tag{6}$$

while $\mu_{\sigma_{0,ij}^{S+N}}$ is computed using Equation (7), which reads

$$\mu_{\sigma_{0,ij}^{S+N}} = \sigma_{0,egg,j}\left(\frac{1 + SNR_{egg,j}}{SNR_{egg,j}}\right) + \Delta_{ij} \tag{7}$$

$$\Delta_{ij} = E[N4DS(U, \theta_{ij}, \phi) - N4DS(U, \theta_{egg,j}, \phi)]_{\phi}$$

In Equation (7), $\Delta_{ij}$ represents the average bias induced by the variation in the incidence angle throughout the entire range of $\phi$.

## 3. Results and Discussion

### 3.1. Comparison between $K_p^{EMP}$ and $K_p^{MED}$ over Sea

The circles in each plot of Figure 4 represent the median of the values $K_p$ ($K_p^{MED}$) provided in the QuikSCAT full-resolution files with respect to the slice index, for each of the four flavors *pol; view*, namely: HH-Pol, aft and HH-Pol, fore (HHA and HHF); VV-Pol, aft and VV-Pol; fore (VVA and VVF), and the total number of samples is reported by flavor. Instead, the crosses represent the estimates ($K_p^{EMP}$) obtained with the methodology proposed in this study. The 68% (95%) confidence intervals (c.i.) of the $K_p$ values are indicated with the solid (dashed) lines. The wind speed values that correspond approximately to the five $\sigma_0$ levels used for the estimates are reported in the caption. These results refer to all orbits of the 10 April 2007 , which are limited to sea measurements within $\pm 60°$ of latitude, for the sake of avoiding any ice contamination. Some remarks follow: (a) the lower limit of the level of noise is higher than $\approx 30\%$, but for low-to-mid wind speed, the level of noise is quite substantial, at least for HH-Pol acquisitions; (b) the dispersion of $K_p$ is very high at low-to-mid wind regimes, especially for HH-Pol outer acquisitions, suggesting that the precision of these measurements is not good in these cases; (c) the level of noise decreases with increasing wind speed and from outer to inner slices, as expected; (d) HH-Pol acquisitions are noisier than VV-Pol. Note that given the same wind speed, $\sigma_0^{HH}$ is lower than $\sigma_0^{VV}$. When $\sigma_0^{HH}$ levels are comparable to $\sigma_0^{VV}$ levels, the noise level is similar; (e) HH-Pol fore acquisitions are noisier than HH-Pol aft; the reason is not yet clear; (f) HH-Pol acquisitions with the indices 6 and 7 (see Figure 2) are noisier than the symmetric indices 0 and 1; neither is this reason clear; (g) the HH-Pol

acquisitions with indices 6 and 7 are outside the 68% c.i.; for mid-to-high $\sigma_0$ levels, they are even outside the 95% c.i., indicating that the differences between $K_p^{EMP}$, estimated from the data, and $K_p^{MED}$, estimated from the product $K_p$s, are significant and that $K_p$ levels are largely underestimated; (h) inner VV-Pol acquisitions have lower levels of noise than reported in the files, while the opposite happens for the outer ones; (i) finally, the statistics for the entire set of 14 orbits are rather similar to those obtained for a single orbit and for the entire set of 14 orbits dated 10 of October 2007, confirming that these results are statistically sound and are not seasonally dependent (not shown). This method can be successfully applied to SeaWinds measurements, given that the dataset is larger than about 5000 samples, for which the accuracy of the estimates will be of few % units. It can also be applied to other scatterometers with a similar architecture (e.g., those onboard Oceansat-2, Scatsat-1, and the HY-2 series), but the minimum number of samples should be evaluated case by case.

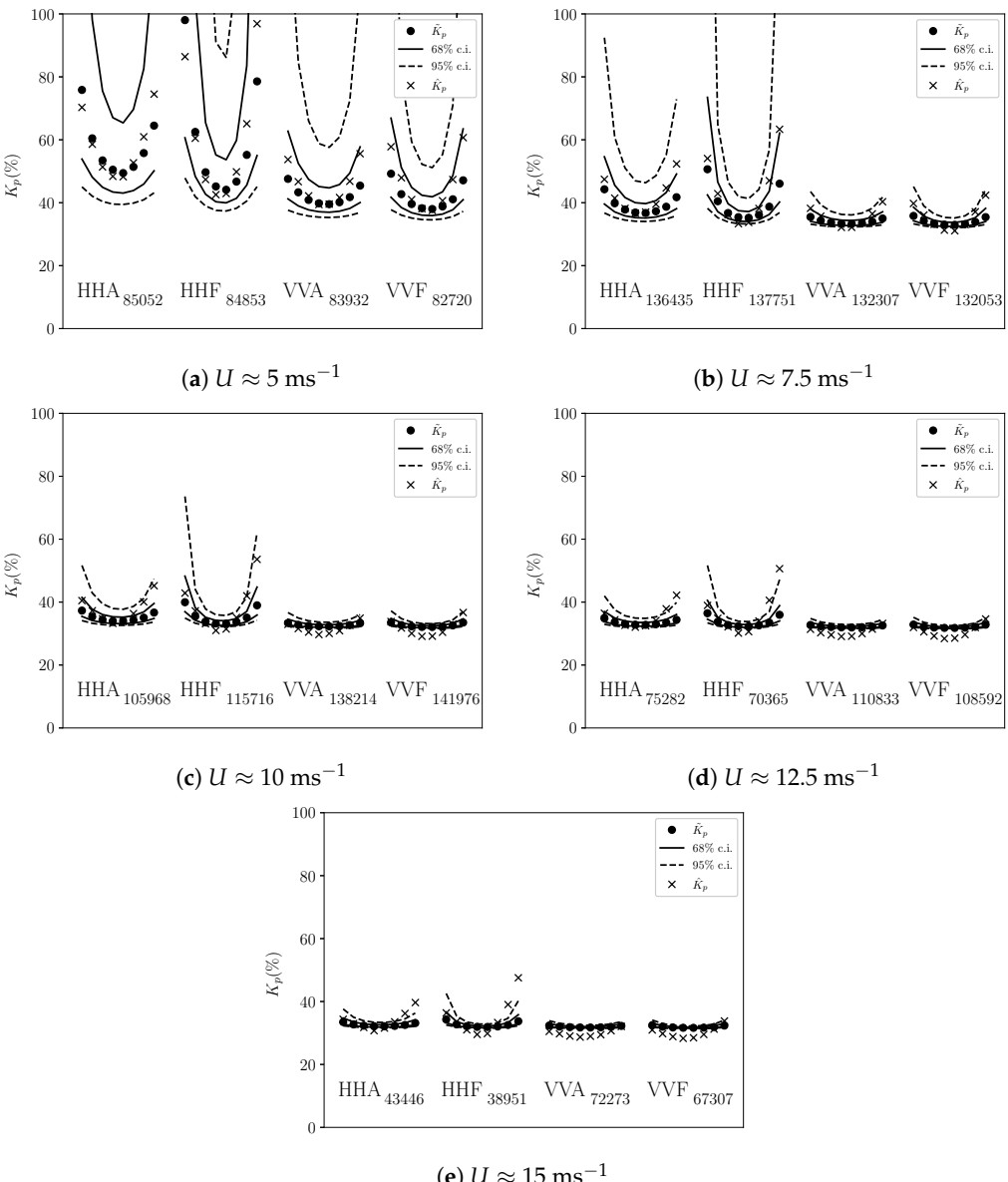

**Figure 4.** In each plot, the circles represent the median of product $K_p$s ($K_p^{MED}$) as a function of the slice index, for each of the four flavors *(pol; view)*, and for the $\sigma_0$ level corresponding approximately to the wind speed indicated in the caption. The 68% (95%) confidence intervals of $K_p$ are represented with solid (dashed) lines. Estimates of $K_p$ from data ($K_p^{EMP}$) are depicted with crosses. The total number of samples is reported by the flavor string, which is composed in the following way: HH (VV) stands for HH-Pol (VV-Pol) and A (F) stands for aft (fore).

### 3.2. Impact of θ-Induced Intra-Egg Biases on $K_p^{EMP}$

The solid lines (dashed) in Figure 5 show the average (WCS) intra-egg biases induced by the variation in the angle of incidence for a wind speed regime of $\approx 10$ ms$^{-1}$. The trend is rather similar for other wind regimes (not shown). It is apparent that the trend is rather linear; therefore, we expect that inner and outer slices will compensate for each other during the slice integration procedure (completed for wind retrieval purposes), at least in the open ocean. In fact, during this procedure, all acquisitions with the same flavor *(pol; view)* are aggregated into a single integrated value. The four integrated values are then used to retrieve the wind vector field. In addition, the biases are larger for the HH-Pol acquisitions and for the aft view. This is in agreement with the values of the expected deviations of the incidence angles of the slice with respect to the incidence angle of the egg, as reported in Table 2. Note that the standard deviations of the deviations follow the same trend as their expected values.

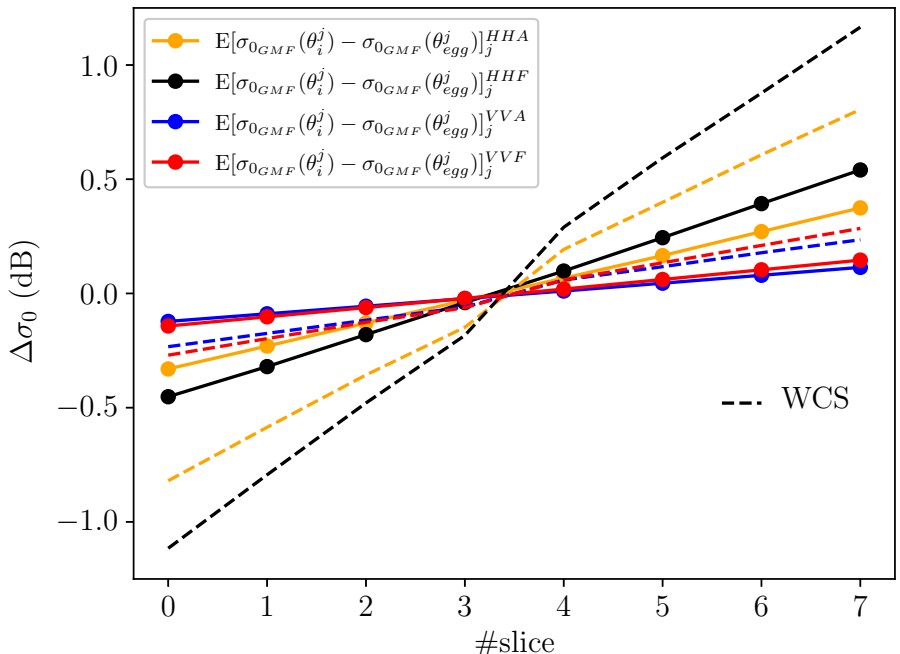

**Figure 5.** Solid (dashed) lines: average (maximum) intra-egg $\sigma_0$ bias in logarithmic units (dB) induced by the incidence angle variations as a function of the slice index, for the four pol-view flavors, and for the $\sigma_0$ level corresponding to 10 ms$^{-1}$. The estimates refer to the orbit with number 40,651. WCS stands for Worst Case Scenario, while the 3-character strings in the legend represent the following: HH (VV) stands for HH-Pol (VV-Pol) and A (F) stands for aft (fore). $i$ in the legend represents the slice index, while $j$ represents the sample index.

**Table 2.** Expected values of the deviations of the incidence angles of the slice with respect to the incidence angle of the egg in degrees $\pm$ their standard deviations for the orbit with number 40,651.

| Slice # | HHA | HHF | VVA | VVF |
|---|---|---|---|---|
| 0 | $0.68 \pm 0.09$ | $0.95 \pm 0.15$ | $0.56 \pm 0.08$ | $0.67 \pm 0.09$ |
| 1 | $0.48 \pm 0.09$ | $0.66 \pm 0.13$ | $0.40 \pm 0.08$ | $0.48 \pm 0.09$ |
| 2 | $0.26 \pm 0.09$ | $0.37 \pm 0.12$ | $0.24 \pm 0.08$ | $0.29 \pm 0.08$ |
| 3 | $0.05 \pm 0.09$ | $0.07 \pm 0.11$ | $0.08 \pm 0.08$ | $0.09 \pm 0.08$ |
| 4 | $-0.16 \pm 0.09$ | $-0.23 \pm 0.12$ | $-0.08 \pm 0.08$ | $-0.11 \pm 0.08$ |
| 5 | $-0.38 \pm 0.09$ | $-0.54 \pm 0.13$ | $-0.24 \pm 0.08$ | $-0.31 \pm 0.08$ |
| 6 | $-0.60 \pm 0.09$ | $-0.85 \pm 0.16$ | $-0.41 \pm 0.08$ | $-0.51 \pm 0.08$ |
| 7 | $-0.82 \pm 0.10$ | $-1.17 \pm 0.19$ | $-0.57 \pm 0.08$ | $-0.72 \pm 0.10$ |

Figure 6 shows the impact of the intra-egg biases on $K_p^{EMP}$ in the WCS at $\approx 15$ ms$^{-1}$. The results for the other regimes are similar; therefore, they are not shown for the sake of brevity. All values are less than 7% and are higher for HH-Pol outer acquisitions. Note that these figures refer to the WCSs. Even under these strict conditions, the differences between $K_p^{EMP}$, estimated from the data, and $K_p^{MED}$, estimated from the product $K_p$s, are meaningful; therefore, the conclusions indicated at points (f), (g) and (i) of Section 3.1 are still valid.

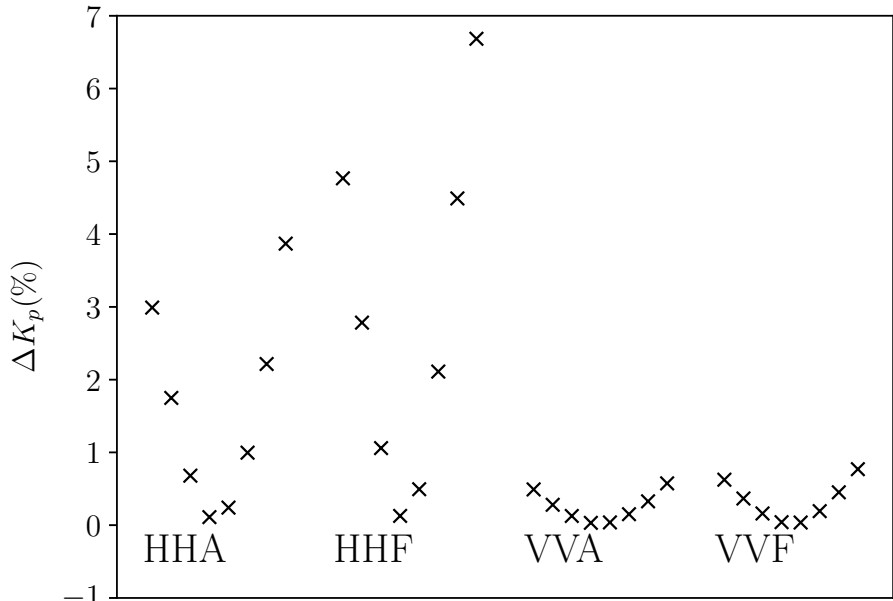

**Figure 6.** Impact of the intra-egg biases on $K_p^{EMP}$ as a function of the slice index for each of the four pol-view flavors and for the $\sigma_0$ level corresponding approximately to 15 ms$^{-1}$ in the WCS. The estimates refer to the orbit with number 40,651. The 3-character strings in the panel represent the following: HH (VV) stands for HH-Pol (VV-Pol) and A (F) stands for aft (fore).

*3.3. Sensitivity to the Type of Surface*

The red (blue) circles in Figure 7 represent $K_p^{MED}$ over the sea (every type of surface), while the red (blue) crosses represent $K_p^{EMP}$ for a wind speed regime of $\approx 15$ ms$^{-1}$. Note that the red symbols are exactly the same as those depicted in Figure 4e and that the blue circles are not visible because they underlie the red. A couple of observations to note: (a) product $K_p$s does not make any distinction in surface type. This is expected because the analytical model used to compute $K_p$ takes into account the thermal noise effect and the fading effect, but it does not consider the variability of the scene, which influences the noise level [12]; (b) $K_p^{EMP}$ evaluated on all types of surfaces is higher than $K_p^{EMP}$ evaluated over the sea. This is expected, since the variability of the scene, which is higher for non-homogeneous surfaces, is expected to increase $K_p$. The differences at low wind regimes are not as evident as at mid and high regimes (not shown).

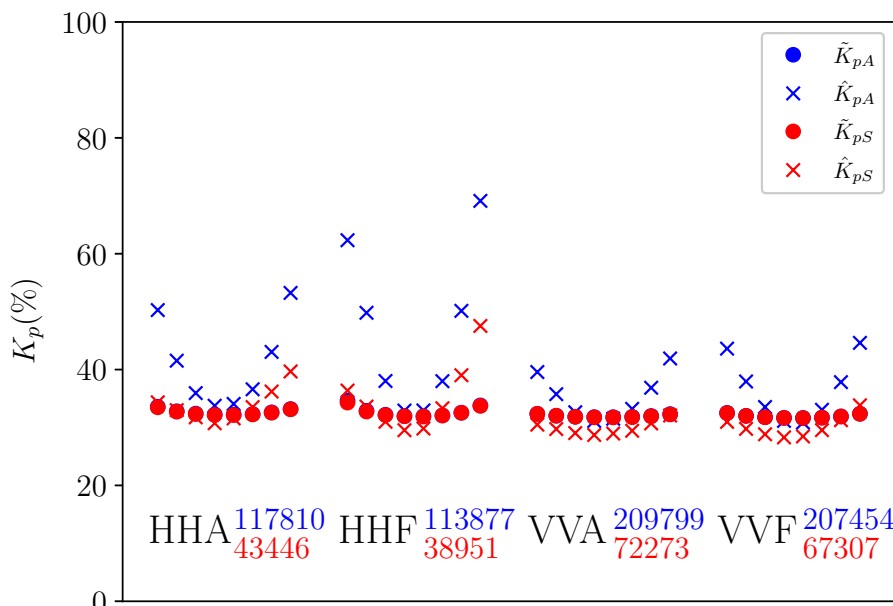

**Figure 7.** The red (blue) circles represent the median of product $K_p$s ($K_p^{MED}$) on sea (all types of surfaces) as a function of the slice index for each of the four flavors *(pol; view)*, and for the $\sigma_0$ level corresponding approximately to 15 ms$^{-1}$. Estimates of $K_p$ ($K_p^{EMP}$) are depicted with crosses with the same color code. The total number of samples is reported by the flavor with the same color code. The 3-character strings in the panel represent the following: HH (VV) stands for HH-Pol (VV-Pol) and A (F) stands for aft (fore).

### 3.4. Inter-Slice Biases and Trends with Azimuth Angle

Figure 8 shows the inter-slice biases of $\sigma_0$ for the orbit with number 40,651. Both figures have the same color bar limits to facilitate comparison, and the limits are symmetric to easily distinguish negative from positive biases. It is apparent that the biases are larger for HH-Pol acquisitions, and they are approximately twice as large as for VV-Pol. This is expected, because the intra-egg variation of the incidence angle is greater for HH-Pol acquisitions (see Table 2). The largest biases are $\approx 0.8$ dB, which is not negligible. Note that these biases may lead to undesired biases in the retrieved winds; therefore, they should be removed. However, their trend appears to be rather linear with the slice index (the distance between the slices is rather constant). They are expected to compensate during the retrieval procedure. However, in coastal areas, the removal of highly contaminated slices can lead to residual biases during the integration procedure prior to the retrieval stage. In light of that, it may be advisable to inter-calibrate them before the integration procedure. This may be completed with an "octuple collocation", which is a generalization of the well-known triple collocation [17]. This method has already been successfully tested for quadruple and quintuple collocation [18,19]. This is left for the future.

Figure 9 shows the trend of the biases with respect to $\sigma_{0,egg}$ for HH-Pol and VV-Pol acquisitions, which are evaluated over all orbits dated 10 April 2007. Single-orbit plots have similar trends but are much noisier. Both figures share the same y-axis limits to make them easily comparable. Once again, the HH-Pol biases are greater than the VV-Pol biases, and their magnitude is comparable to Figure 8, as expected. The trends with azimuth angle are not flat, as would be required, and they are similar for all slices but for the sign. In both the HH-Pol and VV-Pol cases, the biases are greater (in absolute value) for azimuth angles within 0° and 180° than for the complementary interval. The reason for these trends is expected to be related to the global climate zones imprint [20]. Hence, their occurrence probably has no instrumental origin.

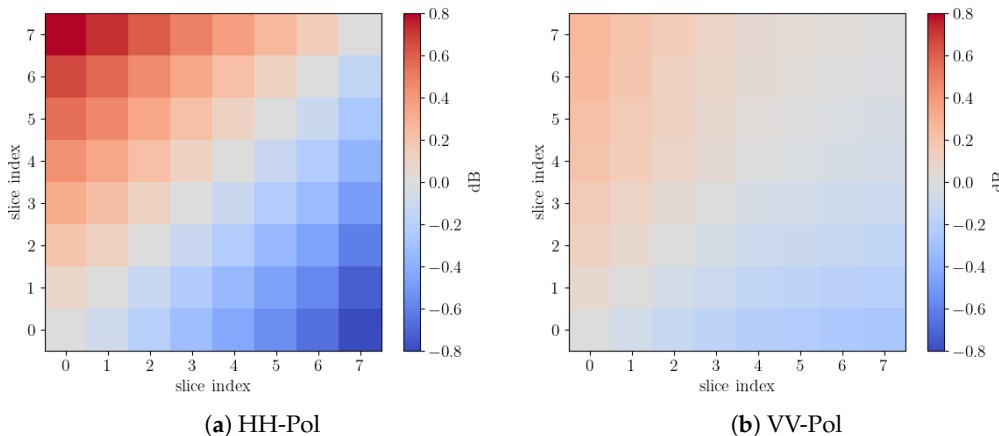

(**a**) HH-Pol                                                (**b**) VV-Pol

**Figure 8.** Left (right): inter-slice biases for HH-Pol (VV-Pol) measurements in the open ocean of the orbit with number 40,651. The matrix is anti-symmetric (change of sign) with respect to the main diagonal.

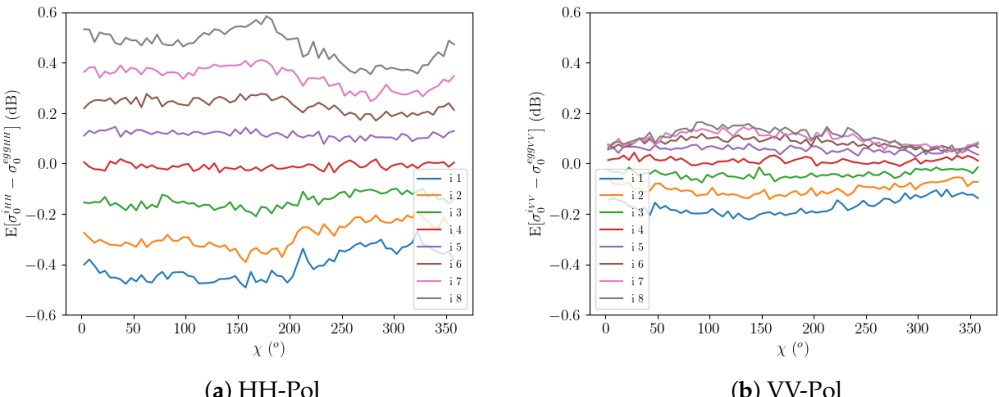

(**a**) HH-Pol                                                (**b**) VV-Pol

**Figure 9.** Left (right): $\sigma_0^{HH-Pol}$ ($\sigma_0^{VV-Pol}$) bias with respect to $\sigma_{0,egg}$ as a function of the azimuth angle in the open ocean, evaluated over all the orbits dated 10 April 2007 . In the y-axis label, HH (VV) stands for HH-Pol (VV-Pol) and *i* in the legend represents the slice index.

### 3.5. Simulation of $\sigma_0$ Distribution

Figure 10 shows the normalized histogram of the real $\sigma_0$ measurements (solid line) compared to those simulated with the model described in Section 2.2.3, for two different cases, the details of which are reported in the panels. The dashed (dotted) line of each graph shows the simulated histogram obtained with $K_p^{EMP}$ ($K_p^{MED}$) evaluated throughout the orbit with number 40,651, while the dashed–dotted line represents the histogram of the simulated $\sigma_0$s, each with its product $K_p$. Figure 10a shows a case where $K_p^{EMP}$ and $K_p^{MED}$ have a very similar value. Note that since $SNR$ is high ($U = 10$ ms$^{-1}$), there are no negative $\sigma_0$s. This case shows that the model described in Equation (4) and the tool described in Equation (5) prove to be effective. Furthermore, the use of a single representative value of $K_p$ is adequate to reproduce the real $\sigma_0$ distribution, at least when the dispersion of $K_p$s is small. Figure 10b shows the case for which the differences between $K_p^{EMP}$ and $K_p^{MED}$ are relevant (see Figure 4e, (HH-Pol; fore) slice number 7). It is apparent that the curve simulated with $K_p^{EMP}$ is wider than the real one, while those obtained with $K_p^{MED}$ and product $K_p$s are narrower. This exercise has been repeated with $K_p^{EMP}$ reduced by $\approx$5% (10% relative to the original value), and the curves overlap rather well (not shown), suggesting that $K_p^{EMP}$ is slightly overestimated in this case, while $K_p^{MED}$ is largely underestimated ($\approx$12% in absolute units).

Figure 11a shows a very noisy case, that for (HH-Pol, fore), slice number 7 of Figure 4a. The style code is identical to that of Figure 10. It is apparent that all simulated curves are much narrower than the real distribution, suggesting that $K_p$s are largely underestimated.

To prove this, Figure 11b shows the same case as Figure 11a, but $K_p$s (and so $K_p^{MED}$) and $K_p^{EMP}$ are multiplied by 1.7 and 1.4, respectively. The curves now overlap quite well, showing once again that this tool is useful for validation purposes and, more important, that HH-Pol acquisitions are very noisy, especially the outer ones and acquired when the antenna looks forward. That HH-Pol fore are noisier than HH-Pol aft acquisitions may be related to Figure 9 and the sampling of climate zones [20]. These results suggest that outer HH-Pol fore acquisitions should be handled with care, considering also the opportunity to discard them in low-wind retrievals. Note that even in such very noisy cases, one single representative value of $K_p$ is sufficient to adequately reproduce the distribution of real $\sigma_0$s.

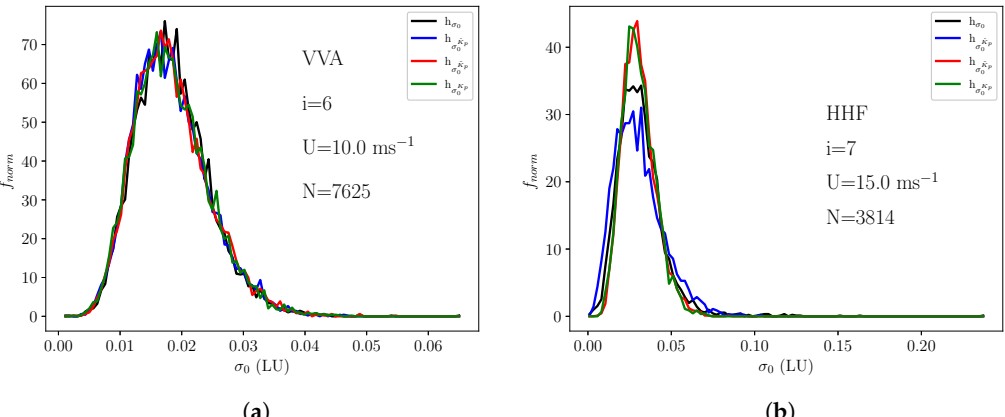

(a)　　　　　　　　　　　　　　(b)

**Figure 10.** In each plot, the solid curve represents the normalized histogram of the real $\sigma_0$s in linear units (LU) of the orbit with number 40,651, while the dashed (dotted) line represents the normalized histogram of the $\sigma_0$s simulated using $K_p^{EMP}$ ($K_p^{MED}$) in the random generator described in Equation (5)). Finally, the dashed–dotted curve shows the simulated $\sigma_0$s obtained using product $K_p$s. Some additional information is reported in each panel, describing the case under examination: the flavor *(pol; view)*, the slice index, the reference wind speed value and the total number of samples, respectively. The flavor string is composed as follows: HH (VV) stands for HH-Pol (VV-Pol) and A (F) stands for aft (fore).

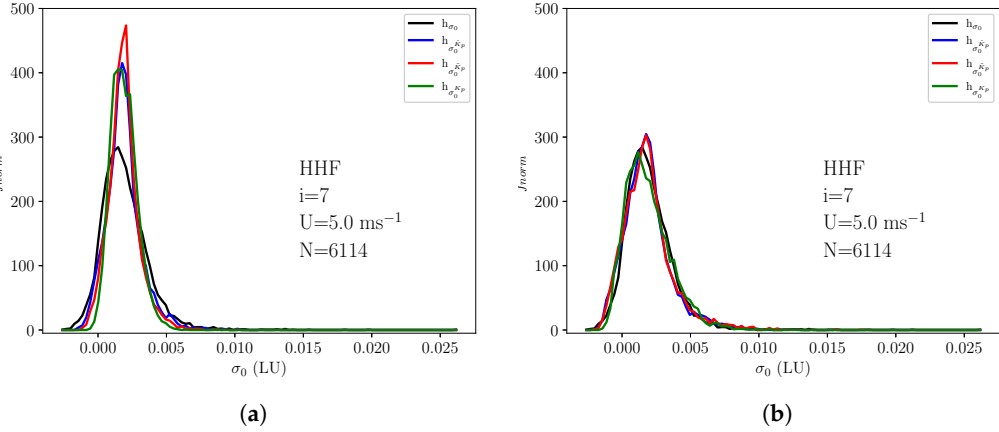

(a)　　　　　　　　　　　　　　(b)

**Figure 11.** In each plot, the solid curve represents the normalized histogram of the real $\sigma_0$s in linear units (LU) of the orbit with number 40,651, while the dashed (dotted) line represents the normalized histogram of the $\sigma_0$s simulated using $K_p^{EMP}$ ($K_p^{MED}$) in the random generator described in Equation (5). Finally, the dashed–dotted curve shows the simulated $\sigma_0$s obtained using product $K_p$s. Some additional information is reported in each panel, describing the case under examination: the flavor *(pol; view)*, the slice index, the reference wind speed value and the total number of samples, respectively. The flavor string is composed as follows: HH (VV) stands for HH-Pol (VV-Pol) and A (F) stands for aft (fore). In the right plot, $K_p$s (and so $K_p^{MED}$) and $K_p^{EMP}$ are multiplied by 1.70 and 1.40, respectively.

## 4. Conclusions

This paper presents a data-driven methodology to estimate the normalized standard deviation ($K_p^{EMP}$) of the noise that affects SeaWinds $\sigma_0$s. This methodology can also be applied to all pencil-beam scatterometers with an architecture similar to SeaWinds. $K_p^{EMP}$ has been compared with the median of product $K_p$s ($K_p^{MED}$) of all orbits dated 10 April 2007. Furthermore, the sensitivities of these estimates to the types of surfaces (sea or any other types) and to the wind regimes have been analyzed. The presence of any biases related to the intra-egg variations of the incidence angle and of the antenna azimuth angle has also been checked. Finally, $K_p^{EMP}$s have been validated using a theoretical model of the $\sigma_0$ distribution.

The results show that SeaWinds $\sigma_0$s may be very noisy, with HH-Pol acquisitions being noisier than VV-Pol, and outer slices (with respect to the egg centroids path) being noisier than inner ones. Note that when $\sigma_0$ levels are comparable, the noise that affects HH-Pol acquisitions is comparable to that of VV-Pol. All these findings are in line with expectations and the literature, even if the exact figures were not known, especially at low levels of $\sigma_0$. In fact, product $K_p$s for outer HH-Pol slices at low levels of $\sigma_0$ can be much higher than 100%, with a very high dispersion, and $K_p^{EMP}$ and $K_p^{MED}$ can reach 100%. With such high $K_p$s, the decision to use such measurements in the retrieval process or discard them is not trivial. Furthermore, the precision of $K_p$ seems to be very poor under such conditions. The answer to this question is left for future studies.

$K_p^{EMP}$s are larger for any kind of surface than for sea, while $K_p^{MED}$s do not show any sensitivity to the type of surface sounded. This result is also expected. In fact, product $K_p$s are computed using a model that takes into account the fading effect and the thermal noise without considering the scene variability effect, while the methodology presented here intrinsically does.

The results show that $\sigma_0$s are shifted due to the intra-egg variation of the incidence angle, and this may lead to undesired biases in the wind retrievals. These shifts are described by the GMF and present a linear trend with the distance between the slices in the footprint. For this reason, they are expected to be accounted for during the retrieval procedure. However, in coastal areas, the regression of slice acquisitions for land decontamination may lead to residual biases. For this reason, to aid land correction, a slice inter-calibration procedure based on multiple collocation may help remove them. This step is planned for the future. The biases found also somewhat depend on the azimuth angle of the antenna. This trend is expected, as the global wind distribution is anisotropic. As before, this needs to be accounted for in a multiple-collocation procedure to avoid overcorrection [20]. These biases can affect $K_p^{EMP}$, but their impact is estimated to be very low: always lower than 7%. The distribution model proposed in this paper proves to be effective in simulating real $\sigma_0$s. Products $K_p$s, $K_p^{EMP}$ and $K_p^{MED}$ have been used in the model for validation purposes. From these comparisons, it is found that $K_p^{EMP}$ is better than $K_p^{MED}$ and product $K_p$s in simulating the real $\sigma_0$ distribution, where some differences are sometimes present, especially in low-wind regimes. In addition, a unique parameter is sufficient to adequately reproduce the distribution of real $\sigma_0$s. The opportunity of using $K_p^{EMP}$ instead of the product $K_p$s in the retrieval stage will be investigated in the future. This study is carried out in the context of the Visiting Science Activity of the Ocean Sea Ice Satellite Application Facilities (OSI SAF) of the European Agency for the Exploitation of Meteorological Satellites (EUMETSAT). The final objective of this activity is to provide users with a dataset of high spatial resolution coastal winds derived from SeaWinds that spans the duration of the mission. In light of this, the next steps are the retrieval of the winds and their validation.

**Author Contributions:** Conceptualization, G.G., A.S. and M.P.; methodology, G.G., A.S. and M.P.; software, G.G., A.V. and J.V.; validation, G.G.; formal analysis, G.G.; investigation, G.G.; resources, G.G. and M.P.; data curation, G.G.; writing—original draft preparation, G.G.; writing—review and editing, A.S., A.V., J.V. and M.P.; visualization, G.G.; supervision, A.S. and M.P.; project administration, G.G., A.S. and M.P.; funding acquisition, A.S., M.P. and G.G. All authors have read and agreed to the published version of the manuscript.

**Funding:** This work was supported in part by the European Organization for the Exploitation of Meteorological Satellites Ocean and Sea Ice Satellite Application Facility (EUMETSAT OSI-SAF) projects under reference VS20_01, VS20_03, and VS21_03, and in part by project INTERACT (PID2020-114623RB-C31), which is funded by MCIN/AEI/10.13039/501100011033.

**Data Availability Statement:** The data used in the paper are available from the PODAAC website with url: https://podaac-opendap.jpl.nasa.gov/opendap/, accessed on 30 September 2022.

**Acknowledgments:** The authors would like to thank David Long from the Brigham Young University and Bryan Stiles and Roy Scott Dunbar from the NASA Jet Propulsion Laboratory for their precious support.

**Conflicts of Interest:** The authors declare no conflict of interest.

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
