# Peer review of "Analysis of Data-Derived SeaWinds Normalized Radar Cross-Section Noise"

_remotesensing, doi:10.3390/rs14215444_

Round 1

Reviewer 1 Report

[1] This article presents a statisticaly method to characterize the bias of RCS data in SeaWinds.

[2] The bias can be useful in practice to retrieve the wind pattern on ocean surface.

[3] Please briefly describe how this bias is used to retrieve the wind pattern.

[4] The symbols used are very confusing, please improve the complexity of symbols.

[5] The writing style must be improved, professional editing service is recommended.

[6] Please prepare a panoramic schematic to clarify the geometrical implication of symbols, like \theta, \phi, wind direction, aft, fore, and so on.

[7] Figure 9: Please use different colors for curves.

[8] The purpose of simulation sigma in Section 2.2.3 is not well justified, please improve.

Reviewer 2 Report

See my comments in the attached document.

Reviewer 3 Report

This paper presents a data-driven methodology to estimate the normalized standard deviation of the noise that affects SeaWinds normalized radar cross section. The article will certainly be useful to a reader who directly develop methods for determining wind from satellite data and is practically useless to readers who are simply consumers of wind data due to its terminological complexity and abundance of technical details.

The comprehensibility of the article to the general public can be improved by explaining in a few sentences why noise information is fundamental (line 49) and how the expected values ​​of the normalized radar cross section are determined (line 53).

Reviewer 4 Report

Specific comments are attached.

Reviewer 5 Report

The manuscript “Analysis of data-derived SeaWinds Normalized Radar Cross Section Noise” by Grieco et al. presents the discussion of the normalized standard deviation of the noise that affects scatterometer normalized Radar Cross Sections in the coastal wind retrieval. The paper is well written, the methodology is clearly explained, and the results are well discussed. Below are minor comments that can help reading the paper.

- Throughout the paper, please pay attention to acronyms that are often not introduced correctly (e.g., ADEOS II, NSCAT, PODAAC, ASCAT, FR, etc.)

- Line 38: the paper talks about "coastal product" but the product itself is never specified. It is assumed that the authors are talking about wind intensity but it would be appropriate to explicitly cite it the first time it is introduced. Furthermore, the altitude to which the wind refers is never defined, is it only in correspondence with the sea surface (0 s.a.s.l.)?

- Line 50: please define the "backscattered normalized radar cross sections"

- Line 66: when do we talk about “low-wind regimes”?

- Lines 90-91: define "number of frames", "number of pulses", and "number of slices"

- Lines 104-105: what do the 0-3 bits of frame_inst_status mean?

- Line 143: “associated with a set of measurements” what is the set of measurements?

- Lines 164-167: Figure 1 shows the position of the centroids for southern Italy. How does this position vary with respect to latitude and longitude? Are there any corrections to be introduced to take into account the Earth's curvature?

- Figure 1: move the legends of the figure. Also, why is the satellite track not aligned with the centroids of the EGG?

- Lines 195-198: since you are using a library already part of python, it is not necessary to specify it and specify the details of the random generator package

Round 2

Reviewer 1 Report

Previous comments have been addressed.

Reviewer 4 Report

The revised draft is a major improvement over the original manuscript. So this paper may be published without any revision.